# Endocrine-Disrupting Chemicals, Hypothalamic Inflammation and Reproductive Outcomes: A Review of the Literature

**DOI:** 10.3390/ijms252111344

**Published:** 2024-10-22

**Authors:** Galateia Stathori, Kyriaki Hatziagapiou, George Mastorakos, Nikolaos F. Vlahos, Evangelia Charmandari, Georgios Valsamakis

**Affiliations:** 1Center for Prevention and Management of Overweight and Obesity, Division of Endocrinology, Metabolism and Diabetes, First Department of Pediatrics, Medical School, National and Kapodistrian University of Athens, ‘Aghia Sophia’ Children’s Hospital, 11527 Athens, Greece; g.stathor@gmail.com (G.S.); evangelia.charmandari@googlemail.com (E.C.); 2Division of Endocrinology, Metabolism and Diabetes, ENDO-ERN Center for Rare Pediatric Endocrine Disorders, First Department of Pediatrics, Medical School, National and Kapodistrian University of Athens, ‘Aghia Sophia’ Children’s Hospital, 11527 Athens, Greece; khatziag@med.uoa.gr; 3Department of Physiotherapy, School of Health and Care Sciences, University of West Attica, 12243 Egaleo, Greece; 4Second Department of Obstetrics and Gynecology, Medical School, National and Kapodistrian University of Athens, ‘Aretaieion’ University Hospital, 11528 Athens, Greece; mastorakg@gmail.com (G.M.); gynoffice04@gmail.com (N.F.V.); 5Division of Endocrinology and Metabolism, Center of Clinical, Experimental Surgery and Translational Research, Biomedical Research Foundation of the Academy of Athens, 11527 Athens, Greece

**Keywords:** endocrine-disrupting chemicals, hypothalamic inflammation, reproduction, TCDD, PCB, TBT, phthalates, BPA, CPF

## Abstract

Endocrine-disrupting chemicals (EDCs) are environmental and industrial agents that interfere with hormonal functions. EDC exposure is linked to various endocrine diseases, especially in reproduction, although the mechanisms remain unclear and effects vary among individuals. Neuroinflammation, particularly hypothalamic inflammation, is an emerging research area with implications for endocrine-related diseases like obesity. The hypothalamus plays a crucial role in regulating reproduction, and its inflammation can adversely affect reproductive health. EDCs can cross the blood–brain barrier, potentially causing hypothalamic inflammation and disrupting the reproductive axis. This review examines the existing literature on EDC-mediated hypothalamic inflammation. Our findings suggest that exposure to 2,3,7,8-tetrachloro-dibenzo-p-dioxin (TCDD), polychlorinated biphenyl (PCB), tributyltin (TBT), phthalates, bisphenol A (BPA), and chlorpyrifos (CPF) in animals is linked to hypothalamic inflammation, specifically affecting the hypothalamic centers of the gonadotropic axis. To our knowledge, this is the first comprehensive review on this topic, indicating hypothalamic inflammation as a possible mediator between EDC exposure and reproductive dysfunction. Further human studies are needed to develop effective prevention and treatment strategies against EDC exposure.

## 1. Introduction

Endocrine disruptors, also known as endocrine-disrupting chemicals (EDCs), are chemical substances—whether synthetic or naturally occurring—that disturb the normal functioning of the endocrine system by altering hormone production, action, or elimination [1]. EDCs pose significant challenges to human and environmental health, being prevalent in nearly every aspect of modern life, including industrial manufacturing, agriculture, and everyday consumer products [2]. Emerging evidence suggests that EDCs have profound effects on development, reproduction, metabolism, and other crucial physiological processes, although often indirectly [3]. Despite their widespread presence, the full extent of their impact and mechanisms of action remains not well understood due to practical obstacles making the establishment of direct causal links between specific EDC exposure and clinical outcomes very difficult [3]. However, after extensive research, it has become clear that there is substantial heterogeneity among the effects of individual EDCs, with outcomes heavily influenced by the age of exposure for each subject [3]. Furthermore, there exists a temporal lag between EDC exposure and the onset of clinical outcomes. Fetal exposure to EDCs may have consequences that are not apparent until adulthood [3]. Moreover, interactions between different types of EDCs may act synergistically, exacerbating their effects. EDC-induced impacts might also extend across generations through epigenetic mechanisms. Scientific evidence shows that fetal EDC exposure can lead to germ cell defects transferable to subsequent generations [4]. These complexities indicate intricate causal pathways, making it difficult to understand the direct mechanisms of action.

Extensive evidence demonstrates that EDCs exert their disruptive effects across various levels of the physiological endocrine axes. The central neuroendocrine systems, primarily represented by the hypothalamus, are not immune to the influence of EDCs. Indeed, research indicates that EDCs can impact the hypothalamus by altering the expression of gonadotropin-releasing hormone (GnRH), a crucial regulator of the reproductive axis [5,6]. However, not only the hypothalamus but the entire central nervous system (CNS) appears to be vulnerable to the deleterious effects of these substances [7]. This assertion derives from published data demonstrating the presence of EDCs within the brain of both animals and humans [8,9,10,11]. EDCs such as parabens, phthalates, bisphenols, polychlorinated biphenyls, dioxins, benzophenones, and organotins have been classified as neurotoxic molecules due to their apparent impact on brain development and promotion of neurodegeneration [12]. Moreover, exposure to these EDCs has been associated with increased damage of blood–brain barrier (BBB) function and neuroinflammation, resulting in neuronal apoptosis and astrogliosis [13]. Interestingly, recent research has associated EDC exposure with behavioral problems, autism, and ADHD [14,15].

Neuroinflammation of the CNS is a complex reaction to injury in cerebral tissue, characterized by the activation of microglia, astrocytes, and immune cells from the bloodstream [16]. This response can be triggered by microbial components, brain trauma, or toxins and leads to the production and secretion of proinflammatory cytokines, macrophages and reactive oxygen species locally, contributing to the inflammatory cascade within the brain [16]. Early neuroinflammation after CNS injury can be protective and prevent further neuronal damage by eliminating harmful substances [17]. However, chronic neuroinflammation is harmful for the CNS and has been associated with neurodegenerative diseases such as Alzheimer’s disease and Parkinson’s disease, as well as psychiatric disorders, including depression and also sleep disorders [18,19,20]. Moreover, evidence suggests that neuroinflammation can impact neuroendocrine centers within the brain, notably the hypothalamus [21]. While molecules classified as EDCs are known to disrupt normal endocrine function and have been shown to induce inflammation within the CNS, their precise role and mechanism in contributing to neuroinflammation remain inadequately explored.

Most of the acquired knowledge regarding the mechanisms and implications of neuroinflammation throughout the CNS originates from animal and in vitro studies. The evolution of imaging techniques has advanced research on neuroinflammation, providing the scientific community with new insights into the molecular mechanisms underlying this complex phenomenon. Functional magnetic resonance imaging (fMRI) and positron emission tomography (PET) scans offer valuable tools for visualizing both structural and functional changes associated with neuroinflammation in humans [22,23]. This scientific advancement has shed light on new aspects of neuroinflammations’ implications in the endocrine field. For instance, emerging evidence suggests a link between hypothalamic inflammation and conditions such as polycystic ovary syndrome (PCOS), obesity, and diabetes [24,25]. The complete spectrum of the etiological pathogenesis and the complications associated with the phenomenon of neuroinflammation remain to be fully elucidated.

As previously noted, exposure to EDCs arising from various industries, including agriculture, construction, cosmetics, and nutrition, has been associated with impaired endocrine function, mainly concerning reproduction and metabolism, and also with non-endocrine outcomes such as psychiatric disorders. Despite extensive research, the exact mechanisms underlying these associations remain unclear, complicating prevention and treatment efforts. Neuroinflammation emerges as a novel probable mediator between EDC exposure and endocrine dysfunction. This review aims to explore the potential impact of EDCs on the development of neuroinflammation within the neuroendocrine sites of the CNS and to assess whether neuroinflammation could serve as a pathogenic mechanism underlying both endocrine and non-endocrine clinical outcomes following EDC exposure. To our knowledge, this is the first comprehensive review assessing this topic.

## 2. EDCs—General Aspects

The idea that certain chemicals used in animal feedlots could enter the human body and affect hormonal activity originated in 1958 [26]. Following this, researchers began linking specific chemicals to rare cancers and reproductive effects in both humans and animals. To date, over 1000 chemicals have been identified as endocrine-disrupting chemicals (EDCs), exhibiting significant diversity among the substances [27]. In its initial scientific statement on EDCs in 2009, the Endocrine Society classified EDCs into industrial solvents/lubricants and their byproducts, plastics, plasticizers, pesticides, fungicides, pharmaceutical agents, and natural chemicals [3] (p. 1). We intend to utilize this same classification framework for our current systematic review.

In addition to the aforementioned diversity among the various substances classified as EDCs, there is also considerable heterogeneity in the mechanisms through which EDCs exert their endocrine-disrupting effects. Evidence suggests that EDCs may act on various sites of the endocrine axes. For instance, dioxin has been shown to mimic estrogen activity by modulating estrogen receptor signaling, while polychlorinated biphenyls (PCBs) block human estrogen sulfotransferase, the enzyme responsible for estrogen inactivation, leading to increased estradiol bioavailability in target tissues [28,29]. Furthermore, data indicate that atrazine increases estrogen concentrations by stimulating aromatase activity [30]. Scientific research has demonstrated that EDCs may act through nuclear and nonnuclear steroid hormone receptors, nonsteroid receptors, enzymatic pathways involved in steroid biosynthesis, and numerous other mechanisms related to endocrine activity [3] (p. 1).

Emerging evidence indicates that among all endocrine axes, the reproductive and thyroid axes are particularly susceptible to the effects of endocrine-disrupting chemicals (EDCs) [3] (p. 1). Most clinical outcomes associated with EDC exposure stem from the reproductive system (e.g., infertility) or thyroid function (e.g., hypothyroidism) [3] (p. 1). PCBs, for instance, have been shown to decrease T4 concentrations in animals, while in rats, BPA exposure has been associated with thyroid resistance syndrome [3]. Within the CNS, the hypothalamus and pituitary gland have been shown to serve as sites of action for these compounds. The hypothalamus hosts gonadotropin-releasing hormone (GnRH) neurons, which regulate the release of gonadotropins from the pituitary gland [31]. These gonadotropins in turn regulate the production of steroid hormones from the gonads, which are essential for normal reproductive function [31]. Additionally, the hypothalamus hosts thyrotropin-releasing hormone (TRH) neurons, which regulate the secretion of thyroid-stimulating hormone (TSH) from the pituitary, thereby influencing thyroid hormone production [32]. Animal and in vitro studies have demonstrated that EDCs can modulate GnRH neuron function, either by enhancing or inhibiting it [33]. For instance, Gore observed that in vitro, methoxychlor and chlorpyrifos, two pesticides, stimulated GnRH gene expression and secretion [5] (p. 2). Klenke et al. found that bisphenol A (BPA) reduces GnRH neuronal activity, potentially through a direct effect on GnRH neurons, independently of estrogen receptors [34]. Moreover, certain EDCs have been shown to impact TRH neuron function. Chronic exposure to tetrabromo-bisphenol A and tributyltin in gestating dams, or acute exposure in newborns, has been linked to heightened TRH activation [35]. In humans, paraben exposure is associated with elevated levels of TSH, whereas in rodents, exposure to parabens leads to a reduction in TSH levels [36]. Despite accumulating evidence of EDC interaction with CNS endocrine centers, the precise mechanisms underlying these effects are not well elucidated. Table 1 presents the available data on reproductive disorders linked to EDC exposure.

## 3. Neuroinflammation and EDCs

Neuroinflammation is a multifaceted biological reaction within the CNS, triggered by various stimuli such as physical agents, traumatic injury, and toxins [16] (p. 2). Initially serving a protective function against further neuronal damage, prolonged or excessive inflammation can negatively impact neuronal integrity and function [16] (p. 2). Recent research on neuroinflammation has revealed previously unknown triggers of this biological process within the CNS. Environmental and lifestyle factors have been implicated in the onset of neuroinflammation, leading to adverse clinical outcomes. For instance, high-fat diet-induced hypothalamic inflammation, observed in both animal models and humans, has been linked to obesity, suggesting a potential role of hypothalamic inflammation as an etiological factor in this complex disease [65,66].

The exact molecular mechanisms underlying the onset of neuroinflammation remain incompletely understood, with variability likely depending on the specific triggers initiating the process. Regardless, there is scientific consensus regarding the significant role of microglia in initiating neuroinflammation. Microglial cells play a crucial role in maintaining the integrity and functionality of CNS tissue throughout life, and along with astrocytes and oligodendrocytes form the non-neuronal components that support the CNS environment [67]. In response to external stimuli, microglia undergo a process often termed “activation” or “polarization”, which involves migration or extension of processes toward the site of damage, cell proliferation, phagocytosis, and activation of astrocytes, accompanied by the production of proinflammatory markers [67,68]. The molecular mechanisms underlying microglia-mediated astrocytosis, which subsequently triggers neuroinflammatory cascades, represent an active field of research. Several signaling pathways have been identified, including activation of the NF-κB pathway, the MAPK pathway, and the JAK/STAT pathway [69,70,71]. Activation of these pathways leads to the transcription of apoptosis-related genes and the release of cytokines, growth factors, and acute-phase response factors, ultimately resulting in neuronal injury [68]. Different stimuli initiating neuroinflammation activate distinct molecular pathways, and each brain region exhibits selective sensitivity to these pathways [68].

Recent evidence suggests a potential association between exposure to EDCs and the development of chronic neuroinflammation, due to disruption of normal immune function within the brain [72]. Animal studies have demonstrated that certain EDCs, such as organotins, can cross the BBB and induce oxidative stress and neurodegenerative processes within the CNS [73]. Specifically, trimethyltin has been shown to activate microglia, increase the expression of glial fibrillary acidic protein (a marker of increased gliosis), and promote astrocyte activation within the rat brain [74]. Moreover, maternal exposure to bisphenol A has been linked to microglial activation and upregulation of genes encoding proinflammatory factors in the cerebral cortex of mice [75]. Consequently, the actions of EDCs within the CNS are associated with neuronal cell degeneration and the onset of neuroinflammation. This neuronal damage may contribute to the clinical endocrine and non-endocrine outcomes that have long been associated with EDC exposure. Table 2 presents the available data on neuroinflammatory effects linked to EDC exposure.

## 4. Industrial Solvents/Lubricants-Dioxin

Dioxins are toxic byproducts of numerous manufacturing processes, but they can also derive from natural phenomena like fires or volcanic eruptions [92]. Common industrial activities producing dioxins include pesticide manufacturing, chlorine bleaching of paper pulp, and incineration [92]. 2,3,7,8-tetrachloro-dibenzo-p-dioxin (TCDD) is the chemical name for dioxin, although often the term “dioxins” encompasses those structurally similar to dioxin chemicals, such as the polychlorinated dibenzo para dioxins (PCDDs) and the polychlorinated dibenzofurans (PCDFs) [93]. Among these substances, TCDD exhibits the highest toxicity [93].

In humans, short-term exposure to high concentrations of dioxins is associated with chloracne, liver dysfunction, neurological impairments, and respiratory deficiencies. Conversely, long-term exposure has been linked to endocrine disruption, cancer, and disruptions in both the nervous and immune systems [93,94]. Concerning reproductive system endocrine disruption, TCDD has been found to inhibit ovulation in immature hypophysectomized rats independently of ovarian steroidogenesis alterations. The authors suggest that this effect is related to follicular rupture, though the exact mechanism remains unclear [37]. Research by Shi et al. demonstrated that TCDD delayed puberty and accelerated the cessation of normal reproductive cycles in female rats without impacting follicular reserves [38]. In the same study, TCDD exposure resulted in a dose-dependent reduction in serum estradiol concentrations, while interestingly serum FSH and LH concentrations, along with their responsiveness to GnRH, were unaffected [38]. In humans, TCDD exposure has been shown to impact semen quality, with an intriguing correlation emerging: exposure before puberty is associated with decreased semen quality, while exposure after puberty is linked to improved semen quality, particularly concerning sperm count and motility [39]. The mechanism behind this paradoxical correlation remains unknown. However, it has been demonstrated that TCDD exposure, whether before or after puberty, leads to decreased estradiol concentrations and increased FSH concentrations, which may account for the reduced semen quality [39].

There is evidence suggesting that TCDD may cause neuronal damage and contribute to the disruption of reproductive neuroendocrine systems. Initially, Huang et al. observed a significant increase in aryl hydrocarbon receptor (AHR) protein expression, the primary receptor through which TCDD exerts its toxicity, in the pituitary cells of mice exposed to TCDD [95]. This finding implies that the pituitary gland is directly affected by TCDD. Additionally, the same research team demonstrated that TCDD induced apoptotic and necrotic cell death in mouse pituitary cells, as evidenced by morphological and biochemical changes indicative of cell death [76]. These findings are consistent with previous reports of pituitary atrophy following TCDD exposure [77,96]. Furthermore, Takeda et al. illustrated that administering TCDD to pregnant rats led to reduced expression of luteinizing hormone (LH) in the pituitaries of their fetuses, along with decreased expression of ovarian and testicular proteins involved in steroidogenesis [97]. To elucidate the underlying mechanism behind this TCDD-mediated effect, the researchers analyzed alterations in the metabolomic profile of the pituitary and hypothalamus in fetuses exposed to TCDD [78]. They discovered that TCDD altered the metabolomic profile in both the pituitary and hypothalamus, with more profound changes observed in the hypothalamus [78]. Specifically, TCDD affected the concentrations of components of the tricarboxylic acid (TCA) cycle in the hypothalamus, leading to the accumulation of oxaloacetic acid, isocitric acid, alpha-ketoglutaric acid, and N-acetylaspartic acid, as well as a decrease in hypothalamic ATP [78]. These alterations not only impact local energy production but also involve TCA components and their products, which serve as neurotransmitters interacting with GnRH neurons in the hypothalamus [78]. The authors suggested that TCDD-mediated hypothalamic damage, coupled with disruptions in the normal metabolomic profile and energy status, could interfere with neurotransmission pathways regulating GnRH neurons, consequently reducing gonadotropin production by the pituitary gland [78].

## 5. Polychlorinated Biphenyls (PCBs)

Polychlorinated biphenyls (PCBs) are a class of synthetic organic compounds that have gained significant attention due to their widespread use and detrimental environmental impacts [98]. Because of their high thermal and chemical stability, PCBs found extensive application in industrial processes, electrical equipment, and consumer products throughout the 20th century [98]. However, their persistence, bioaccumulative nature, and toxicity led to their classification as persistent organic pollutants (POPs) and a ban on their production in the USA in 1978 [99]. Nevertheless, due to novel sources of PCBs, such as unintentionally generated contaminants found in certain pigments and varnishes, a significant quantity of traditionally manufactured PCBs persists globally [100].

Human exposure to PCBs primarily occurs through contaminated food sources, notably fish, as well as through inhalation of air and ingestion of water [100]. High levels of PCB exposure have been linked to various health conditions, including cancer, cardiovascular issues, and neurocognitive dysfunction [101]. In the endocrine field, PCBs have been linked to repercussions on the thyroid and the reproductive system [101,102]. They appear to exhibit estrogenic activity, potentially by hindering estrogen sulfotransferase, thereby enhancing estrogen bioavailability [43]. Perinatal exposure to PCBs in female mice has been correlated with reduced concentrations of pre-ovulatory luteinizing hormone (LH) and progesterone, along with diminished ovarian and uterine weights during ovulation [40]. Furthermore, prenatal exposure to PBC in rats has been associated with altered sexual behavior, negatively affecting the chances of mating and reproduction [41,42]. In humans, PCB exposure has been positively correlated with decreased sperm motility [44]. A cohort study of workers exposed to PCBs found a positive association between PCB exposure and prostate cancer mortality, with a strong exposure–response relationship [45]. Moreover, prenatal PCB exposure in male humans has been linked to delayed puberty: Grandjean et al. studied 438 adolescent boys from a birth cohort in the Faroe Islands, known for elevated PCB exposure. They measured PCBs in cord blood at age 14, finding that higher prenatal PCB exposure was correlated with lower serum levels of LH and testosterone. Additionally, SHBG was positively associated with both prenatal and concurrent PCB exposure. These results suggest that PCB-associated delayed puberty may stem from a central hypothalamo-pituitary mechanism [46].

Scientific evidence suggests that PCBs can disrupt the reproductive neuroendocrine systems. In a study conducted by Gore et al., the effects of two different PCB mixtures (Aroclor 1221 and Aroclor 1254) on hypothalamic cells were investigated in vivo [6] (p. 2). Aroclor 1221 was found to stimulate the expression of GnRH, leading to increased levels of GnRH peptide, whereas high doses of Aroclor 1254 inhibited GnRH expression [6] (p. 2). The authors proposed that these findings indicate a central hypothalamic origin for PCB exposure-related endocrine disruption of reproductive function [6] (p. 2). Another study demonstrated that PCB exposure in Atlantic croaker fish led to a reduction in GnRH neurons within the hypothalamus, accompanied by decreased GnRH receptors in the pituitary and diminished responsiveness of luteinizing hormone (LH) to GnRH [103]. The precise mechanism by which PCBs interfere with central neuroendocrine sites and alter their function remains unclear. However, Bell et al. suggested that prenatal exposure to PCBs may induce neuroimmune disruption and hypothalamic inflammation [79]. Specifically, their study examined the brains of one-day-old offspring of rats exposed to PCBs during gestation [79]. Rats prenatally exposed to PCBs exhibited increased expression of inflammatory genes such as Ikbkb, Ccl22, and Tnf within the hypothalamus compared to control rats [79]. Lkbkb encodes a kinase subunit (IKKβ) that promotes cytokine production and an inflammatory response, Ccl22 is associated with macrophage activation, and TNF (encoded by the Tnf gene) acts as a cytokine within the brain and can promote neuronal damage and apoptosis [79]. Consequently, the alteration of inflammatory gene expression within the hypothalamus due to PCB exposure could promote local neuroinflammation, leading to neuroendocrine disruption and contributing to the adverse clinical outcomes associated with PCB exposure.

## 6. Organotins—Tributyltin (TBT)

Tributyltin (TBT) is an organotin compound historically utilized as a biocide in marine anti-fouling paints, primarily to inhibit the growth of algae and other organisms on ship hulls [104]. Additionally, TBT finds applications as a wood preservative, industrial water disinfectant, and agricultural fungicide [104]. The primary sources of TBT entering the environment are contaminated water and sediments from harbors, largely due to its extensive use in boat paints [105]. Marine organisms are directly exposed to TBT through water and sediment contact, while terrestrial organisms, including humans, may encounter exposure through ingestion of contaminated food, water, and soil, as well as via the application of biocidal products and atmospheric deposition [105].

In terms of its endocrine-disrupting effects, the existing literature predominantly associates TBT exposure with alterations in metabolic profile and the reproductive system. Rodent studies have linked TBT exposure to increased adipogenesis and obesity, attributed to the activation of peroxisome proliferator-activated receptor gamma (PPARγ) receptors, pivotal regulators of adipogenesis [106,107,108]. Fish and gastropods exposed to TBT have exhibited the development of masculine sex characteristics in female organisms, termed imposex development [47]. Podratz et al. demonstrated that female rats exposed to TBT exhibited irregular menstrual cycles, reduced duration of reproductive cycles, decreased levels of estradiol in the blood, decreased ovarian weight, and elevated progesterone levels [48]. Furthermore, Sena et al. revealed that TBT-exposed rats experienced dysfunction in the gonadotropic axis, characterized by reduced surge in luteinizing hormone (LH) levels, decreased expression of gonadotropin-releasing hormone (GnRH), and diminished responsiveness to kisspeptin, accompanied by increased body weight, hyperinsulinemia, and hyperleptinemia [49]. Given the role of leptin in regulating the gonadotropic axis, the authors proposed that TBT-induced disruption of this axis might involve impaired signaling of leptin, kisspeptin, and GnRH in the hypothalamus [49].

Evidence suggests that TBT exposure may induce neuronal damage in the brain. Studies in rats have revealed that TBT exposure disrupts the blood–brain barrier (BBB), leading to oxidative damage, astrocyte activation, and increased levels of inflammatory molecules such as interleukin 6 (IL-6) and nuclear factor kappa B (NF-κB), ultimately resulting in neurodegeneration [80]. Further investigations have demonstrated that TBT exposure induces reactive oxygen species (ROS) generation and neuronal apoptosis in various brain regions, notably the hypothalamus, hippocampus, cerebellum, and striatum [81]. Additionally, TBT-exposed female rats have exhibited elevated expression of inducible nitric oxide synthase (iNOS) and inflammation in the hypothalamus, along with increased inflammatory cells and neuronal apoptosis in the pituitary gland [82]. These findings suggest a potential central origin of TBT’s endocrine-disrupting activity, mediated by hypothalamic inflammation and disruption of hypothalamic–pituitary axes [82]. TBT has been mostly associated with obesity generation and reproductive dysfunction. Given the hypothalamus’s role in regulating appetite satiety and hosting GnRH neurons, hypothalamic dysfunction resulting from local neuroinflammation may contribute to the adverse clinical outcomes associated with TBT exposure, particularly obesity and reproductive dysfunction.

## 7. Plasticizers—Phthalates

Phthalates represent a class of synthetic chemicals widely utilized in industrial and consumer products as plasticizers [109]. These compounds, characterized by their ability to enhance the flexibility and durability of plastics, find extensive application in various sectors, including the manufacturing of PVC plastics, adhesives, cosmetics, and personal care products [109]. There are many phthalate types, such as diethylhexyl phthalate (DEHP), dibutyl phthalate (DBP), diethyl phthalate (DEP), diisononyl phthalate (DiNP), and diisodecyl phthalate (DiDP), but DEHP and its metabolite mono(2-ethylhexyl) phthalate (MEHP) are the most frequently used [110]. Despite their widespread use, concerns have arisen regarding the potential adverse effects of phthalate exposure on both animal and human health [109].

Humans can be exposed to phthalates through water, food, dermal contact by plastic products, or even by breathing polluted air, particularly for those living in industrial areas [111]. While there were initial concerns about phthalates causing cancer, this link is now disputed [112]. However, research has shown that phthalate exposure in humans is linked to obesity, insulin resistance, and diabetes [113]. Moreover, phthalates are known for their endocrine-disrupting properties, mostly impacting the reproductive system. Human exposure to phthalates has been correlated with decreased plasma testosterone levels, impaired semen quality, precocious breast development in girls, precocious puberty, and endometriosis [50,51,52,53,114]. Despite the recognized associations, the precise mechanisms underlying these reproductive effects remain elusive. Notably, research suggests that phthalates may induce central precocious puberty by upregulating kisspeptin levels, subsequently activating gonadotropin-releasing hormone (GnRH) neurons in the hypothalamus, as evidenced by higher urinary phthalate metabolite levels and elevated serum kisspeptin concentrations in girls with precocious puberty compared to control, age-matched girls [54].

Evidence suggests that phthalates may induce neuronal damage in the brain, which could contribute to the adverse endocrine effects associated with phthalate exposure. First, Shwe et al. demonstrated that phthalate exposure induces neuroinflammation in the hypothalamus of mice [83]. Their study on male mice exposed to phthalates revealed increased expression of inflammatory cytokines (IL-1β and TNF-α), elevated oxidative stress markers, and enhanced microglia activation in the hypothalamus, indicative of neuroinflammation [83]. Given the hypothalamus’s role in regulating appetite and puberty onset, these findings are particularly relevant to understanding the association of phthalate exposure with central precocious puberty and obesity. Similarly, another research group showed that phthalate exposure in mice resulted in neurotoxicity in the cerebral cortex, striatum, and brainstem, accompanied by elevated IL-1β and TNF-α levels, disturbances in oxidative status, and increased pro-apoptotic proteins within these brain regions [84]. Consistently with these findings, Zhou et al. demonstrated that phthalates led to neuron atrophy, increased microglia activation markers, and neuronal apoptosis in the hippocampus of mice, along with a significant increase in brain-derived neurotrophic factor (BDNF) levels in the same brain area [85]. BDNF, released by activated microglia during neuroinflammation, has been shown to regulate the gonadotropic axis in sheep by stimulating gonadotropin-releasing hormone (GnRH) expression in the hypothalamus and promoting gonadotropin secretion in the pituitary [115,116]. Furthermore, women with functional hypothalamic amenorrhea exhibit significantly lower plasma BDNF concentrations compared to eumenorrheic control women [117]. These findings suggest that the reproductive effects associated with phthalate exposure may be attributed to neuroinflammation in the reproductive centers of the central nervous system, possibly mediated by BDNF stimulation.

## 8. Plastics—Bisphenol A

Bisphenol A (BPA) is a synthetic compound used primarily in the production of polycarbonate plastics and epoxy resins, a class of reactive polymers [118]. These materials are commonly found in food and beverage containers, water bottles, thermal paper receipts, medical devices, and dental sealants [118]. From these products, BPA can contaminate food, water, and the environment, leading to widespread exposure [118]. Contamination in humans can occur through direct contact with BPA-containing products or indirectly through environmental contamination from manufacturing processes [119].

BPA exposure in humans has been associated with impaired neurological function and an increased risk of chronic medical conditions such as obesity, diabetes, and cardiovascular diseases [118]. In the endocrine field, BPA is known to interact with estrogen receptors (ERs) due to its estrogenic properties, although its relative affinity for both ERα and ERβ is up to 10,000 times lower than that of estradiol [120,121]. Studies have shown that in pregnant rodents, BPA crosses the placenta and binds to α-fetoprotein, resulting in increased BPA bioavailability within the fetal circulation [122]. BPA exposure in mice has been linked to early puberty, reduced vaginal weight, increased mammary gland density, abnormal prostate and urethra development, and increased prostate size [55,56,57,58,59]. The effects of BPA on the reproductive system are mainly attributed to its estrogen-like activity and its affinity for estrogen receptors in estrogen-targeted organs [3]. However, there is scientific evidence that BPA can also interact with hypothalamic reproductive centers. An in vitro study demonstrated that BPA can stimulate GnRH secretion within the hypothalamus of female rats [60]. Another in vitro study revealed that BPA-exposed female and male pubertal rats exhibited increased ER-α expressing neurons within the arcuate nucleus, the ventromedial nucleus, and the medial preoptic area of the hypothalamus [123]. Interestingly, Stoker et al. found that BPA exposure in male rats increases food intake and promotes obesity, while simultaneously enhancing kisspeptin expression, which plays a pivotal role in the activation of GnRH neurons [124]. As both the appetite centers and kisspeptin neurons are located within the hypothalamus, the authors suggested that BPA may alter hypothalamic signals, thereby affecting both feeding behavior and the reproductive axis [124].

Although the main effects of BPA within the reproductive centers of the hypothalamus seem to be ER-mediated, there is scientific evidence that BPA may exert an inflammatory effect in this specific brain area. High-fat diets have been shown to promote an inflammatory response within the hypothalamus of mice, leading to the local release of cytokines and interleukins [66] (p. 4). In this context, Lama et al. studied the effects of BPA in the CNS of mice fed a high-fat diet [86]. The results showed that BPA increased monocyte recruitment and promoted the release of inflammation markers such as IL-1 and TNF within the hypothalamus, exacerbating diet-induced hypothalamic inflammation [86]. Additionally, BPA increased the expression of the Toll-like receptor 4 (TLR4) protein within the CNS of these mice [86]. TLR4 has been shown to play a pivotal role in neuroinflammation by promoting microglia activation and blood–brain barrier (BBB) disruption [86]. Another research team found that BPA-exposed mice presented with hypothalamic inflammation, increased astrocyte activation, and impaired function of proopiomelanocortin (POMC) neurons [88]. Similarly to the findings of Lama et al., these effects were found to be mediated by the TLR4 pathway [88]. In the same context, Salehi et al. demonstrated that BPA exposure resulted in neuroinflammation of POMC neurons in the hypothalamus of mice, as evidenced by the increased concentration of inflammatory markers such as IL-6 within these neurons [87]. From the above evidence, it appears that BPA exposure can promote inflammation within hypothalamic centers. This BPA-mediated hypothalamic inflammation, possibly through the activation of the TLR4 pathway, could play a role in the related clinical endocrine outcomes associated with BPA exposure.

## 9. Pesticides—Chlorpyrifos (CPF)

Chlorpyrifos (CPF) is a widely utilized organophosphate pesticide commonly applied in agriculture on crops such as corn, soybeans, fruit, and nut trees to manage a variety of pests [125]. In use since the 1960s, CPF’s safety has been increasingly investigated, especially following the Gulf War [125]. Pesticide exposure, including CPF, has been significantly linked to Gulf War syndrome in veterans and is considered, alongside psychological factors, one of the most probable causes of this condition. Humans and animals can be exposed to CPF through direct contact during application, consumption of contaminated food, inhalation, and exposure to contaminated water sources [125]. CPF exposure has been associated with various adverse health effects, including neurodevelopmental issues, cardiovascular diseases, and hematological malignancies [125]. CPF inhibits the enzyme acetylcholinesterase, leading to nervous system manifestations such as muscle weakness, seizures, or in severe cases death [125]. Additionally, several studies have demonstrated that CPF acts on various regions of the CNS, disrupting neuronal integrity and function [126].

The literature suggests that CPF’s endocrine-disruptive actions primarily affect the reproductive system, although animal studies indicate potential impacts on thyroid function as well [127]. CPF exposure, along with exposure to other pesticides, has been associated with low semen quality, characterized by decreased sperm concentration and motility [61,62]. Epidemiological studies have also revealed a probable association between pesticide exposure and male urogenital tract disorders such as cryptorchidism and hypospadias [63,64]. Although the mechanism by which CPF exerts its endocrine-disrupting effects is not fully elucidated, it has been shown that CPF can interfere with androgen biosynthesis, disrupting normal steroidogenesis and exerting an antiandrogen effect [128]. Furthermore, CPF has been shown to increase GnRH expression in hypothalamic cells independently of estrogen receptors [5] (p. 2).

Several animal studies indicate that CPF is associated with neuroinflammation. Locker et al. demonstrated that CPF caused inflammation in the cortex and hippocampus of mouse brains, with increased local expression of inflammation markers such as IL-6, IL-1β, TNF, and leukemia-inhibitory factor (LIF) [89]. This neuroinflammation was likely independent of CPF’s cholinergic effects [89]. In another study, researchers using diffusion MRI examined the brains of mice treated with diisopropyl fluorophosphate (DFP), a substance chemically similar to chlorpyrifos and other pesticides [90]. The results showed that DFP exposure led to inflammatory changes in the hypothalamus, thalamus, hippocampus, amygdala, and piriform cortex, with increased expression of inflammatory proteins such as TNFα, IL-6, and IL-1β in these brain areas [90]. Interestingly, corticosterone administration significantly enhanced these DFP-mediated effects [90]. Similarly, O’Callaghan et al. found that DFP-exposed mice exhibited widespread brain neuroinflammation, which was amplified up to 300-fold by corticosterone administration [129]. Furthermore, Adedara et al. demonstrated that CPF-treated rats showed a significant increase in lipid peroxidation (LPO) and a decrease in antioxidant enzymes within the hypothalamus, indicative of oxidative damage [91]. The same alterations were observed in the testes of the same rats. Additionally, CPF-treated mice exhibited decreased levels of LH, FSH, and testosterone [91]. These findings suggest that CPF-mediated reproductive deficits could be caused by both testicular and hypothalamic damage induced by CPF [91].

## 10. Metals and Trace Elements

Cadmium is a naturally occurring metal often found as a byproduct of mining, smelting, and refining zinc, lead, and copper ores. It is commonly found in the environment in soil, air, and water, largely due to industrial processes and the use of phosphate fertilizers [130]. Cadmium exposure can occur through ingestion of contaminated food and water or inhalation of dust and fumes. Toxicity from cadmium can lead to severe health effects, such as kidney damage, bone fragility, and respiratory issues. Chronic exposure is also linked to an increased risk of cancer [130]. Cadmium has been proven to act as an endocrine disruptor mainly targeting the reproductive system [131]. Studies have shown that cadmium exposure can lead to various adverse reproductive health effects. For instance, cadmium exposure has been linked to delayed puberty, alterations in steroidogenesis, pregnancy loss, menstrual cycle disorders, premature births, and reduced birth weights. Additionally, cadmium exposure has been associated with preeclampsia [130].

Lead is a naturally occurring toxic metal found in the Earth’s crust. It is commonly found in the environment due to industrial processes, such as mining, smelting, and the use of leaded gasoline [132]. Exposure to lead can cause severe health effects, including kidney dysfunction, cardiovascular diseases, and increased risk of cancer [132]. Lead is recognized as an endocrine disruptor. Studies have shown that lead exposure can affect thyroid hormone levels and is associated with reproductive health issues, such as delayed puberty, reduced fertility, and adverse pregnancy outcomes [133].

Both cadmium and lead have been shown to exert neurotoxic effects within the CNS [134]. Saedi et al. showed that prepubertal female rats exposed to cadmium exhibited a decreased number of neurons and oligodendrocytes in the arcuate and dorsomedial nuclei of the hypothalamus. Furthermore, these cadmium-exposed rats displayed decreased serum levels of LH and FSH [135]. These findings suggest that cadmium may interfere with the function of the gonadotropic axis through inflammatory processes within the hypothalamus. Additionally, lead exposure has been associated with neuroinflammation via the activation of reactive oxygen species (ROS), leading to neuronal oxidative damage and impaired synaptic transmission [134]. This evidence suggests that neuroinflammation could be implicated in the endocrine-disrupting activity of these toxic elements.

The recognition of the hazardous health effects of metals and trace elements has led to the development of various strategies to combat environmental pollution from these substances. Techniques such as phytoremediation using plants, microbial fermentation, and synthesized nanoparticles have been proposed for this purpose. Specifically, the sunflower (*Helianthus annuus* L.) has been demonstrated to remove cadmium from polluted soil, the bacterium Lactobacillus plantarum has been shown to remove cadmium from rice, and alumina nanoparticles have been shown to extract cadmium from groundwater [136].

## 11. Discussion

In this literature review, we investigated whether exposure to EDCs is associated with neuroinflammation and whether this neuroinflammation could underlie the endocrine-disrupting effects of EDCs, particularly within the reproductive system. EDC exposure has been linked to various adverse reproductive outcomes in both males and females. Proposed mechanisms of action include interference with sex hormone production, interaction with sex hormone receptors, and modulation of receptor expression within the hypothalamus [3] (p. 1). Neuroinflammation, along with its clinical consequences, is a relatively new field of study that could enhance our understanding of EDCs’ impact on the reproductive axis.

Our findings indicate that all examined EDCs are associated with neuroinflammation in various brain regions, including the hypothalamus, as demonstrated by animal studies. Specifically, exposure to TCDD results in cell death in the pituitary gland of mice, alters the metabolomic profile in the pituitary and hypothalamus, affects local energy production, and disrupts neurotransmitter interactions with GnRH neurons in the hypothalamus of exposed fetuses [76,78] (p. 5). Prenatal exposure to PCBs increases the expression of inflammatory genes within the hypothalamus of newborn rats [79] (p. 6). In rats, TBT exposure leads to blood–brain barrier disruption, astrocyte activation, and neurodegeneration in various brain regions, including the hypothalamus. Female rats specifically exhibit hypothalamic inflammation and neuronal apoptosis in the hypothalamus and pituitary [80,82] (p. 7). Phthalate exposure induces hypothalamic neuroinflammation in mice, resulting in increased expression of several inflammatory cytokines and elevated BDNF levels, which are implicated in the regulation of the gonadotropic axis [83,85] (p. 8). BPA promotes neuroinflammation through monocyte recruitment and inflammatory cytokine release within the hypothalamus of mice, increasing TLR4 expression [86,87,88] (p. 9). DFP exposure in mice results in hypothalamic inflammation characterized by increased interleukin expression and oxidative damage, amplified by corticosterone [90,91] (p. 10).

Animal studies have demonstrated that exposure to EDCs leads to differing clinical outcomes between sexes independently of inherent sex differences. For instance, EDC exposure is linked to precocious puberty in female rats and delayed puberty in male rats [3] (p. 1). Our collected animal data indicate that EDCs can induce hypothalamic inflammation in both male and female subjects, suggesting that the hypothalamic effects of EDCs may be sex-specific. Similarly, in humans, EDC exposure is associated with various adverse reproductive effects, with notable sex differences. For example, the literature strongly supports a link between EDC exposure and male infertility, while the association with female infertility is less established [137,138]. EDC-related hypothalamic inflammation, which affects GnRH neurons, may contribute to these adverse outcomes [139]. However, it is possible that males and females exhibit different responsiveness and sensitivity to this neuroinflammation, potentially explaining the varied reproductive outcomes associated with EDC exposure. Further research is needed to determine whether hypothalamic inflammation accounts for the sex-related differences in reproductive outcomes due to EDC exposure.

Another important issue to consider is the possible role of gut microbiota changes in the relationship between EDC exposure, hypothalamic inflammation, and reproductive disorders. EDC exposure has been linked to gut microbiota dysbiosis: human studies have shown that exposure to EDCs, including phthalates and BPA, leads to alterations in the gut microbial community [140]. On the other hand, the gut microbiota has been found to regulate reproductive hormones, though the mechanisms remain unclear. Dysbiotic gut microbiota has been associated with reproductive disorders such as PCOS and endometriosis [141]. Recent research in mice has shown that the gut microbiota regulates hypothalamic inflammation through hypothalamic GLP-1 receptors [142]. Given these data and our findings on the role of EDCs in hypothalamic inflammation, the gut microbiota could be a potential mediator of EDC-related reproductive outcomes via hypothalamic inflammation.

Despite the long-known hazardous health effects of EDC exposure, these chemicals are still widely used in various everyday applications. Notably, many studies included in our review investigated the effects of prenatal EDC exposure, demonstrating that prenatal exposure can promote hypothalamic inflammation similarly to postnatal exposure. This finding aligns with established associations between prenatal exposure and adverse reproductive outcomes, as demonstrated by large epidemiological studies [3] (p. 1). This underscores the importance of advancing scientific knowledge on EDCs to protect human health, particularly that of pregnant women, from EDC exposure, which can lead to early-life inflammation of central endocrine centers.

The association between EDCs and hypothalamic inflammation is gaining interest within the scientific community. Recently, Alshelh et al. provided the first in vivo evidence of neuroinflammation in patients with Gulf War illness, a condition primarily associated with EDC exposure, leading the way for further research in this field [143]. The exact etiology of many reproduction-related medical conditions, such as polycystic ovary syndrome, remains unknown, and the incidence of conditions like central precocious puberty is inexplicably rising [144]. Future research should focus on the potential role of EDC-mediated neuroinflammation in the human reproductive axis and its adverse consequences. It is also important to ascertain whether EDC-induced neuroinflammation has a direct effect on human fertility. This could promote the development of effective prevention and treatment strategies, as well as the establishment of successful national policies for the appropriate use of EDCs.

## 12. Conclusions

This study indicates that EDC exposure is associated with neuroinflammation, specifically targeting the hypothalamic centers of the gonadotropic axis. Our comprehensive review suggests hypothalamic inflammation as a potential mediator between EDC exposure and reproductive dysfunction. Further human studies are essential to elucidate the precise mechanisms and long-term impacts of EDC exposure on reproductive health.

## Figures and Tables

**Table 1 ijms-25-11344-t001:** Disorders of the reproductive system associated with specific EDC exposure in animals and humans.

EDC	Effects in Animals	Effects in Humans
2,3,7,8-tetrachloro-dibenzo-p-dioxin (TCDD)	- Inhibition of ovulation in rats [37] - Delayed puberty in female rats [38] - Reduced serum estradiol concentrations in female rats [38]	- Decreased semen quality [39]
Polychlorinated biphenyl (PCB)	- Decreased concentrations of LH and progesterone in female rats [40] - Altered sexual behavior, reducing reproductive chances in rats [41,42]	- Increased estrogen bioavailability [43] - Decreased semen quality and increased incidence of prostate cancer [44,45] - Delayed puberty, decreased serum LH and testosterone levels in males [46]
Tributyltin (TBT)	- Development of imposex in fish and gastropods [47] - Irregular menstrual cycles, decreased estradiol levels, and increased progesterone levels in female rats [48] - Reduced LH surge and decreased GnRH expression in rats [49]	
Phthalates		- Decreased plasma testosterone levels [50] - Impaired semen quality [51] - Precocious breast development in girls [52] - Endometriosis [53] - Precocious puberty in girls [54]
Bisphenol A (BPA)	- Early puberty in female mice [55] - Reduced vaginal weight in mice [56] - Increased mammary gland density in mice [57] - Abnormal development of the prostate and urethra in male mice [58] - Increased prostate size in mice [59] - In vitro stimulation of GnRH secretion in the hypothalamus of female rats [60]	
Chlorpyrifos (CPF)	- In vitro increase in GnRH expression in hypothalamic cells [5]	- Decreased sperm concentration and motility [61,62] - Cryptorchidism and hypospadias [63,64]

**Table 2 ijms-25-11344-t002:** Neuroinflammatory effects associated with specific EDC exposure.

2,3,7,8-tetrachloro-dibenzo-p-dioxin (TCDD)	- Apoptotic and necrotic cell death in mouse pituitary cells [76] - Pituitary atrophy in female rats [77] - Alteration of the metabolomic profile in the pituitary and hypothalamus of rats [78]
Polychlorinated biphenyl (PCB)	- Increased expression of inflammatory genes (Ikbkb, Ccl22, Tnf) in the hypothalamus of rats [79]
Tributyltin (TBT)	- Disruption of the blood–brain barrier (BBB), oxidative damage, astrocyte activation, increased levels of inflammatory molecules (IL-6, NF-κB), neurodegeneration in rats [80] - ROS generation and neuronal apoptosis in the hypothalamus, hippocampus, cerebellum, and striatum of male rats [81] - Elevated expression of iNOS in the hypothalamus, increased inflammatory cells, neuronal apoptosis in the pituitary gland of female rats [82]
Phthalates	- Increased expression of inflammatory cytokines (IL-1β and TNF-α), elevated oxidative stress markers, enhanced microglia activation in the hypothalamus of male mice [83,84] - Neuron atrophy, increased microglia activation markers, neuronal apoptosis, increased BDNF levels in the hippocampus of mice [85]
Bisphenol A (BPA)	- Increased monocyte recruitment and release of inflammation markers (IL-1 and TNF) in the hypothalamus, elevated TLR4 levels within the CNS of rats [86,87] - Increased astrocyte activation, impaired function of proopiomelanocortin (POMC) neurons in the hypothalamus of mice [88]
Chlorpyrifos (CPF)	- Increased expression of inflammation markers (IL-6, IL-1β, TNF, LIF) in the cortex and hippocampus of mouse brains [89] - Increased expression of inflammatory proteins (TNFα, IL-6, IL-1β) in the hypothalamus, thalamus, hippocampus, amygdala, and piriform cortex of mice [90] - Increase in lipid peroxidation (LPO), decrease in antioxidant enzymes in the hypothalamus of rats [91]

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
