# Peer review of "Endocrine-Disrupting Chemicals, Hypothalamic Inflammation and Reproductive Outcomes: A Review of the Literature"

_ijms, 2024, doi:10.3390/ijms252111344_

Round 1
Reviewer 1 Report
Comments and Suggestions for Authors
Galateia Stathori,et al. did a review work about EDCs and nerve systemic changes in mammals named "Endocrine-disrupting chemicals, Hypothalamic Inflammation and Reproductive outcomes: A review of the literature". Indeed, this is a very important review to elucidate the EDCs adverse effects on the nervous system mainly focused on the neuron in CNS in detail. However, some sentences are still need to be improved. For instance, Line Page67-68, ”Thus, CNS exposure to EDC’s may have non endocrine outcomes as well, with repercussions specific to the exposed brain region.” this description is not correct. In fact, CNS exposure to EDCs is not only to demage the structure and function of brain region and also compromise the HPO and HPT axies, subsequently affecting the sexual hormone secretion.
Author Response
Reviewer 1
Comments of Reviewer 1: Galateia Stathori et al. did a review work about EDCs and nerve systemic changes in mammals named "Endocrine-disrupting chemicals, Hypothalamic Inflammation and Reproductive outcomes: A review of the literature". Indeed, this is a very important review to elucidate the EDCs adverse effects on the nervous system mainly focused on the neuron in CNS in detail. However, some sentences are still need to be improved. For instance, Line Page67-68,”Thus, CNS exposure to EDC’s may have non endocrine outcomes as well, with repercussions specific to the exposed brain region.” this description is not correct. In fact, CNS exposure to EDCs is not only to demage the structure and function of brain region and also compromise the HPO and HPT axis, subsequently affecting the sexual hormone secretion.
Answer to Reviewer 1: We sincerely appreciate your thorough review and your comment. The sentence “Thus, CNS exposure to EDC’s may have non-endocrine outcomes as well, with repercussions specific to the exposed brain region” (Lines: 72-74) has been revised for clarity and subsequently deleted.
Reviewer 2 Report
Comments and Suggestions for Authors
Dear authors,
the article is very interesting and well written.
Here my comments to further improve it
Abstract:
Line 18-31: The abstract is comprehensive, but the last sentence could be more specific in outlining what new insights this review provides. Consider highlighting the novelty of linking EDC exposure to hypothalamic inflammation and reproductive outcomes.
Introduction
Introduction: Line 35-44: The introduction is concise but should emphasize the gap in human data more strongly. Consider adding a brief mention of what remains unknown and why this review is timely.
Line 48-52: The introduction mentions temporal lags and epigenetic effects but does not provide context. Briefly expand on why these phenomena complicate the understanding of EDC effects.
Section 2: EDCs - General Aspects: Line 128-136: When discussing the susceptibility of the thyroid axis, it would be helpful to briefly mention specific EDCs known to affect thyroid function and the associated health outcomes.
Section 4. Line 225-229: The discussion on TCDD and reproductive function could be expanded to include more detailed molecular mechanisms if available. For example, how does TCDD impact the hypothalamic-pituitary-gonadal axis?
Line 283-285: The PCB section effectively details GnRH effects, but the implications for human health need to be more explicitly drawn out.
Consider to add a 11. Conclusion paragraph with an overall summary and future perspectives. (which could also be taken from the last part of the discussion)
Comments on the Quality of English LanguageMinor editing of English language required.
Author Response
Reviewer 2
Comments of Reviewer 2: Dear authors, the article is very interesting and well written.Here my comments to further improve it
Comment No 1. Abstract: Line 18-31: The abstract is comprehensive, but the last sentence could be more specific in outlining what new insights this review provides. Consider highlighting the novelty of linking EDC exposure to hypothalamic inflammation and reproductive outcomes.
Answer to Comment No 1. We thank you for your critical evaluation and constructive comments, which have greatly contributed to the improvement of our manuscript. We changed the last sentence of the abstract: ‘’Our findings indicate that exposure to 2,3,7,8-tetrachloro-dibenzo-p-dioxin (TCDD), Polychlorinated biphenyl (PCB), Tributyltin (TBT), Phthalates, Bisphenol A (BPA), and Chlorpyrifos (CPF) in animals is linked to hypothalamic inflammation, which in turn disrupts the gonadotropic axis. These results suggest that hypothalamic inflammation may mediate the relationship between EDC exposure and reproductive dysfunction. Further human studies are necessary to develop effective prevention and treatment strategies against EDC exposure.’’ To: “Our findings suggest that exposure to 2,3,7,8-tetrachloro-dibenzo-p-dioxin (TCDD), Polychlorinated biphenyl (PCB), Tributyltin (TBT), Phthalates, Bisphenol A (BPA), and Chlorpyrifos (CPF) in animals is linked to hypothalamic inflammation, specifically affecting the hypothalamic centers of the gonadotropic axis. To our knowledge, this is the first comprehensive review on this topic, indicating hypothalamic inflammation as a possible mediator between EDC exposure and reproductive dysfunction. Further human studies are needed to develop effective prevention and treatment strategies against EDC exposure.” (Lines:26-32) in order to emphasize the novelty of our topic.
Comment No 2. Introduction: Line 35-44: The introduction is concise but should emphasize the gap in human data more strongly. Consider adding a brief mention of what remains unknown and why this review is timely.
Answer to Comment No 2. We thank you for this comment. We changed the last part of the Introduction: ‘’As previously noted, exposure to EDC’s arising from various industries including agriculture, construction, cosmetics, and nutrition, has been associated to impaired endocrine function mainly concerning reproduction and metabolism and also to non-endocrine outcomes such as psychiatric disorders. The aim of this review is to explore the potential impact of EDCs on the development of neuroinflammation within the neuroendocrine sites of the CNS, and to assess whether neuroinflammation could serve as a pathogenic mechanism underlying both endocrine and non-endocrine clinical outcomes following exposure to EDCs.’’ to: ‘’As previously noted, exposure to EDC’s arising from various industries including agriculture, construction, cosmetics, and nutrition, has been associated to impaired endocrine function mainly concerning reproduction and metabolism and also to non-endocrine outcomes such as psychiatric disorders. Despite extensive research, the exact mechanisms underlying these associations remain unclear, complicating prevention and treatment efforts. Neuroinflammation emerges as a novel probable mediator between EDC exposure and endocrine dysfunction. This review aims to explore the potential impact of EDCs on the development of neuroinflammation within the neuroendocrine sites of the CNS and to assess whether neuroinflammation could serve as a pathogenic mechanism underlying both endocrine and non-endocrine clinical outcomes following EDC exposure. To our knowledge, this is the first comprehensive review assessing this topic.’’ (Lines:105-112) to highlight the existing knowledge gap.
Comment No 3. Line 48-52: The introduction mentions temporal lags and epigenetic effects but does not provide context. Briefly expand on why these phenomena complicate the understanding of EDC effects.
Answer to Comment No 3.
We appreciate your insightful comment. In response, we have expanded the section from ‘’Furthermore, there exists a temporal lag between EDC exposure and the onset of clinical outcomes indicating complex pathways of causality. Moreover, interactions between different types of EDCs may act synergistically, exacerbating their effects and EDC-induced impacts might extend across generations through epigenetic mechanisms’’ to ‘’Furthermore, there exists a temporal lag between EDC exposure and the onset of clinical outcomes. Fetal exposure to EDCs may have consequences that are not apparent until adulthood. Moreover, interactions between different types of EDCs may act synergistically, exacerbating their effects. EDC-induced impacts might also extend across generations through epigenetic mechanisms. Scientific evidence shows that fetal EDC exposure can lead to germ cell defects transferable to subsequent generations. These complexities indicate intricate causal pathways, making it difficult to understand the direct mechanisms of action.’’ (Lines:49-57)
providing additional information on these phenomena that complicate the understanding of EDC effects.
Comment No 4. Section 2: EDCs - General Aspects: Line 128-136: When discussing the susceptibility of the thyroid axis, it would be helpful to briefly mention specific EDCs known to affect thyroid function and the associated health outcomes.
Answer to Comment No 4. Thank you for your suggestion regarding the discussion of EDCs and thyroid function. We have expanded our text to include examples: ‘’PCBs for instance have been shown to decrease T4 concentrations in animals, while in rats, BPA exposure has been associated to thyroid resistance syndrome’’ (Lines: 138-139). Additionally, we have already referenced the effects of EDCs on thyroid function (Lines 153-156): ‘’Chronic exposure to tetrabromo-bisphenol-A and tributyltin in gestating dams, or acute exposure in newborns, has been linked to heightened TRH activation. In humans, paraben exposure is associated with elevated levels of TSH, whereas in rodents, exposure to parabens leads to a reduction in TSH levels.’’
Comment No 5. Section 4.Line 225-229: The discussion on TCDD and reproductive function could be expanded to include more detailed molecular mechanisms if available. For example, how does TCDD impact the hypothalamic-pituitary-gonadal axis?
Answer to Comment No 5. We appreciate your insightful comment regarding the discussion on TCDD . In response, we have expanded the paragraph from ‘’Concerning reproductive system endocrine disruption, TCDD has been found to inhibit ovulation in immature hypophysectomized rats, an effect unrelated to ovarian steroidogenesis alterations. Research by Shi et al. demonstrated that TCDD delayed puberty and accelerated the cessation of normal reproductive cycles in female rats, without impacting follicular reserves. Notably, in the same study, TCDD exposure led to a dose-dependent reduction in serum estradiol concentrations. In humans, TCDD exposure has been shown to impact semen quality, with an intriguing correlation emerging: exposure before puberty is associated with decreased semen quality, while exposure after puberty is linked to improved semen quality, particularly concerning sperm count and motility’’
To: ‘’Concerning reproductive system endocrine disruption, TCDD has been found to inhibit ovulation in immature hypophysectomized rats, independent of ovarian steroidogenesis alterations. The authors suggest that this effect is related to follicular rupture, though the exact mechanism remains unclear. Research by Shi et al. demonstrated that TCDD delayed puberty and accelerated the cessation of normal reproductive cycles in female rats without impacting follicular reserves. In the same study, TCDD exposure resulted in a dose-dependent reduction in serum estradiol concentrations, while, interestingly, serum FSH and LH concentrations, along with their responsiveness to GnRH, were unaffected. In humans, TCDD exposure has been shown to impact semen quality, with an intriguing correlation emerging: exposure before puberty is associated with decreased semen quality, while exposure after puberty is linked to improved semen quality, particularly concerning sperm count and motility. The mechanism behind this paradoxical correlation remains unknown. However, it has been demonstrated that TCDD exposure, whether before or after puberty, leads to decreased estradiol concentrations and increased FSH concentrations, which may account for the reduced semen quality.’’ (Lines:230-245) providing more detailed information on the potential mechanisms by which TCDD impacts the hypothalamic-pituitary-gonadal axis.
Comment No 6.Line 283-285: The PCB section effectively details GnRH effects, but the implications for human health need to be more explicitly drawn out.
Answer to Comment No 6.:
We appreciate your valuable feedback regarding the PCB section. In response, we have expanded the section : ‘’ In humans, PCB exposure has been positively correlated with decreased semen quality and prostate cancer incidence. Moreover, prenatal PCB exposure in male humans has been linked to delayed puberty’’ To ‘’ In humans, PCB exposure has been positively correlated with decreased sperm motility [69]. A cohort study of workers exposed to PCBs found a positive association between PCB exposure and prostate cancer mortality, with a strong exposure–response relation-ship [70]. Moreover, prenatal PCB exposure in male humans has been linked to delayed puberty: Grandjean et al. studied 438 adolescent boys from a birth cohort in the Faroe Is-lands, known for elevated PCB exposures. They measured PCBs in cord blood and at age 14, finding that higher prenatal PCB exposure correlated with lower serum levels of LH and testosterone. Additionally, SHBG was positively associated with both prenatal and concurrent PCB exposures. These results suggest that PCB associated delayed puberty may stem from a central hypothalamo-pituitary mechanism ‘’ (Lines: 292-302)
to more explicitly draw out the implications for human reproductive function.
Comment No 7. Consider to add a 11. Conclusion paragraph with an overall summary and future perspectives. (which could also be taken from the last part of the discussion)
Answer to Comment No 7.
Thank you for your suggestion regarding the addition of a conclusion paragraph. We have incorporated a new Section (12), which includes an overall summary and outlines future perspectives: ‘’Conclusion: This study indicates that EDC exposure is associated with neuroinflammation, specifically targeting the hypothalamic centers of the gonadotropic axis. Our comprehensive review suggests hypothalamic inflammation as a potential mediator between EDC exposure and reproductive dysfunction. Further human studies are essential to elucidate the precise mechanisms and long-term impacts of EDC exposure on reproductive health’’. (Lines:640-644)
Reviewer 3 Report
Comments and Suggestions for Authors
In the present manuscript “Endocrine-disrupting chemicals, Hypothalamic Inflammation and Reproductive outcomes: A review of the literature”, G. Stathori and coworkers examined existing literature on Endocrine-Disrupting Chemicals (EDC)-mediated hypothalamic inflammation. The authors suggest that hypothalamic inflammation may mediate the relationship between EDC exposure and reproductive dysfunction. However, currently, further human studies are necessary to develop effective prevention and treatment strategies against EDC exposure.
Overall, I think that the manuscript is intriguing and timely. However, I have some suggestion to improve the overall quality of review.
1) Gut microbiota composition is related to EDC-mediated hypothalamic inflammation and reproductive outcomes (Calero-Medina L et al. Sci Total Environ 2023, 886:163991; Fabozzi G. et al. Cells. 2022, 11(21): 3335). Please deeply discuss this very intriguing topic of current research.
2) Please add a subchapter in the revised version of paper on Potentially toxic elements (i.e. Cadmium and Lead). Indeed, these toxic elements impair synaptic transmission and exacerbate neuroinflammation, impacting central nervous system health; their dysregulation has been also implicated in neurodegenerative disorders such as Alzheimer's disease and Parkinson's disease (Doroszkiewicz J et al. Int J Mol Sci. 2023, 24(21):15721).
3) In light of the observation above resumed, please to discuss on the possible application of nutraceutics and/or antioxidants/antinflammatory compounds, that, targeting different molecular pathways and properly combined with good agricultural practice and healthy eating habits, could provide a definite strategy to prevent and counteract Cadmium toxicity, particularly in menopausal period and in different organs. For your convenience you could consider and comment the following references (Marini H.R. et al. Metabolites 2023, 13(6):722; Levine L. and Hall J.E. Climacteric. 2023, 26(3):206-215; Genchi G. et al. Int J Environ Res Public Health. 2020, 17(11):3782 .
4) The authors could add in Graphical form the molecular pathways discussed in the present review; in this way, I feel that the readers can better understand the possible strategies to develop effective prevention and treatment against EDC exposure.
Comments on the Quality of English LanguageMinor editing of English language is required.
Author Response
Reviewer 3
Comments of Reviewer 3: In the present manuscript “Endocrine-disrupting chemicals, Hypothalamic Inflammation and Reproductive outcomes: A review of the literature”, G. Stathori and coworkers examined existing literature on Endocrine-Disrupting Chemicals (EDC)-mediated hypothalamic inflammation. The authors suggest that hypothalamic inflammation may mediate the relationship between EDC exposure and reproductive dysfunction. However, currently, further human studies are necessary to develop effective prevention and treatment strategies against EDC exposure.Overall, I think that the manuscript is intriguing and timely. However, I have some suggestion to improve the overall quality of review.
Comment No 1. Gut microbiota composition is related to EDC-mediated hypothalamic inflammation and reproductive outcomes (Calero-Medina L et al. Sci Total Environ 2023, 886:163991; Fabozzi G. et al. Cells. 2022, 11(21): 3335). Please deeply discuss this very intriguing topic of current research.
Answer to Comment No 1.
Thank you for highlighting this intriguing topic that can significantly enhance the value of our paper.In response, we have added a comprehensive paragraph in the discussion section that references the relationship between gut microbiota composition, EDC-mediated hypothalamic inflammation, and reproductive outcomes. :’’ Another important issue to consider is the possible role of gut microbiota changes in the relationship between EDC exposure, hypothalamic inflammation, and reproductive disorders. EDC exposure has been linked to gut microbiota dysbiosis: Human studies have shown that exposure to EDCs, including phthalates and BPA, leads to alterations in the gut microbial community. On the other hand, the gut microbiota has been found to regulate reproductive hormones, though the mechanisms remain unclear. Dysbiotic gut microbiota has been associated with reproductive disorders such as PCOS and endometriosis. Recent research in mice has shown that the gut microbiota regulates hypothalamic inflammation through hypothalamic GLP-1 receptors .Given this data and our findings on the role of EDCs in hypothalamic inflammation, the gut microbiota could be a potential mediator of EDC-related reproductive outcomes via hypothalamic inflammation.’’ (Lines: 605-616)
Comment No 2. Please add a subchapter in the revised version of paper on Potentially toxic elements (i.e. Cadmium and Lead). Indeed, these toxic elements impair synaptic transmission and exacerbate neuroinflammation, impacting central nervous system health; their dysregulation has been also implicated in neurodegenerative disorders such as Alzheimer's disease and Parkinson's disease (Doroszkiewicz J et al. Int J Mol Sci. 2023, 24(21):15721).
Answer to Comment No 2.
We appreciate your suggestion regarding the inclusion of a subchapter on potentially toxic elements.We consider this addition timely and important. In response, we have added Section 10: ‘’Metals and Trace Elements’’, incorporating six new references (Lines: 520-561 References: 130 -136). This new section focuses on the endocrine-disrupting effects of cadmium and lead, and provides data on their implications for neuroinflammation
Comment No 3. In light of the observation above resumed, please to discuss on the possible application of nutraceutics and/or antioxidants/antinflammatory compounds, that, targeting different molecular pathways and properly combined with good agricultural practice and healthy eating habits, could provide a definite strategy to prevent and counteract Cadmium toxicity, particularly in menopausal period and in different organs. For your convenience you could consider and comment the following references (Marini H.R. et al. Metabolites 2023, 13(6):722; Levine L. and Hall J.E. Climacteric. 2023, 26(3):206-215; Genchi G. et al. Int J Environ Res Public Health. 2020, 17(11):3782 .
Answer to Comment No 3.
Thank you for your insightful suggestion. In response, we have added the following paragraph under the new section on Metals and Trace Elements: ‘’The recognition of the hazardous health effects of metals and trace elements has led to the development of various strategies to combat environmental pollution from these substances. Techniques such as phytoremediation using plants, microbial fermentation, and synthesized nanoparticles have been proposed for this purpose. Specifically, the sun-flower (Helianthus annuus L.) has been demonstrated to remove cadmium from polluted soil, the bacterium Lactobacillus plantarum has been shown to remove cadmium from rice, and alumina nanoparticles have been shown to extract cadmium from groundwater’’. (Lines:554-561)
Comment No 4. The authors could add in Graphical form the molecular pathways discussed in the present review; in this way, I feel that the readers can better understand the possible strategies to develop effective prevention and treatment against EDC exposure.
Answer to Comment No 4. Thank you for your valuable suggestion regarding the addition of a graphical representation of the molecular pathways discussed in our review. While we agree that this would enhance the readers' understanding, we regret to inform you that we currently lack the expertise and experience to create a high-quality and presentable graphical appropriate for the journal's level. We appreciate your understanding and hope that the detailed descriptions in the text will suffice.
Round 2
Reviewer 3 Report
Comments and Suggestions for Authors
Thank you for addressing my comments well. I have no further remarks.